# Effects of Chestnut Hydrolysable Tannin on Intake, Digestibility, Rumen Fermentation, Milk Production and Somatic Cell Count in Crossbred Dairy Cows

**DOI:** 10.3390/vetsci10040269

**Published:** 2023-04-01

**Authors:** Tipwadee Prapaiwong, Wuttikorn Srakaew, Sukanya Poolthajit, Chalong Wachirapakorn, Chaiwat Jarassaeng

**Affiliations:** 1Department of Animal Science, Faculty of Agriculture, Khon Kaen University, Khon Kaen 40002, Thailand; 2Department of Animal Science and Fisheries, Faculty of Science and Technology, Rajamangala University of Technology Lanna Nan, Nan 55000, Thailand; 3Division of Theriogenology, Faculty of Veterinary Medicine, Khon Kaen University, Khon Kaen 40002, Thailand

**Keywords:** chestnut hydrolysable tannin, feed utilization, milk production, somatic cell count, crossbred dairy cows

## Abstract

**Simple Summary:**

Mastitis is a disease that has a significant impact on the global dairy industry, including productivity, quality and farmer income. There are several methods for preventing mastitis. The use of secondary substances (hydrolysable tannins) to prevent the development of mastitis in dairy cows is one example. Hydrolysable tannins have been shown to reduce the bacteria that cause mastitis in the udder. A previous study report discovered that hydrolysable tannins could eliminate both Gram-positive and Gram-negative bacteria that cause mastitis. The aim of this study was to determine how chestnut hydrolysable tannins (CHT) supplemented at levels of 0, 3.15, 6.30 and 9.45 g/day in milking cows affected intake, digestibility, rumen fermentation, and milk yield and quality. As result, CHT supplementation was found to have a promising effect on nutrient apparent digestibility and rumen fermentation. Furthermore, the somatic cell count (SCC) in milk was reduced, indicating lower bacterial contamination in milk.

**Abstract:**

This study was conducted to determine the effects of chestnut hydrolysable tannin (CHT) on intake, digestibility, rumen fermentation, milk yield and somatic cell count in crossbred dairy cows (>75% Holstein Friesian). Four crossbred dairy cows (467.6 ± 35.2 kg BW) were assigned to be supplemented with one of four levels of CHT according to a 4 × 4 Latin square design. Dietary treatments included the control (without CHT supplementation) and CHT treatments that consisted of supplementation with 3.15, 6.30 and 9.45 g CHT/day. Rice straw was given ad libitum. The results showed that increasing levels of CHT tended to quadratically decrease rice straw intake (*p* = 0.06). However, total dry matter intake (DMI) and other nutrients were not different (*p* > 0.05) among the dietary treatments. The apparent digestibility of DM, organic matter (OM) and crude protein (CP) in cows with CHT treatments were higher (*p* < 0.05) than those of control cows. Milk yield and milk composition were not different (*p* > 0.05) among treatments. Lactose yield tended to increase linearly (*p* = 0.09) as CHT supplementation increased. Ruminal pH and ammonia nitrogen (NH_3_-N) were not different (*p* > 0.05), but total volatile fatty acids (VFAs) increased linearly (*p* < 0.05) as CHT levels increased. The somatic cell count (SCC) and somatic cell score (SCS) in the CHT treatments were different (*p* < 0.01) than those in the control treatment. In conclusion, it appears that CHT supplementation improved feed utilization and influenced SCC in crossbred dairy cows. Long-term research is needed to confirm the benefit of CHT supplementation.

## 1. Introduction

There is currently interest in using plant extracts in ruminant diets to improve production in ruminants and animal health. Several plant extracts have been studied in ruminants for their use as rumen fermentation modifiers, antimicrobials and antioxidants, [1,2,3] as well as for therapeutic applications [4]. Tannins are plant extracts that have been widely shown to improve nitrogen utilization and enhance the productivity of ruminants. Tannins are classified into two structures: hydrolysable and condensed tannins. Condensed tannins (CTs) can bind to dietary proteins and reduce their degradability in the rumen [5,6], whereas hydrolysable tannins (HTs) have a lower molecular weight than CTs and are more easily absorbed from the intestine, potentially increasing their toxicity to animals.

Some studies have found no difference between tannin structures on ruminal protein degradation, the digestibility of proteins, or animal performance [7]. One study [8] reported that adding a 0.45% tannin mixture (HTs and CTs at 1:2 ratio) decreased feed efficiency but increased milk protein content. Feeding a tannin extract mixture may reduce environmentally labile urinary N excretion without affecting milk yield but at the expense of decreased feed intake. In addition, chestnut hydrolysable tannin (CHT) has been reported to have positive effects on ruminants, showing better protein utilization, increased weight, wool and milk, increased fertility, and the prevention of parasitic infections as well as improved animal health [9]. Supplementation of CHT at 0.49% in the diet had no effect on dry matter intake and milk yield in dairy cows [10]. Liu et al. [11] reported that CHT extract supplemented at 1% of the diet had no effect on intake, body weight, body condition score, milk yield and milk composition, while decreasing the somatic cell score (SCS). Another study [12] also reported that the addition of CHT at 40 g/day induced a better milk yield from crossbred dairy cows, whereas there was no effect on milk protein, lactose, fat or total solids and a reduced somatic cell count (SCC) in milk. The antibacterial activity of the HT may be responsible for the decrease in SCC in dairy cow’s milk.

Hydrolysable tannins have been reported to exert antibacterial activity by damaging the lipid bilayer membranes and cell membranes of bacteria [13], inhibiting extracellular microbial enzymes and the complexation of metal ions, and depleting substrates [14,15]. Maisak et al. [16] revealed that tannin from chestnut wood has antimicrobial activity against *Streptococcus agalactiae*, which is the most important contagious bacterium causing a high SCC in milk [17]. Furthermore, our preliminary in vitro study of CHT as an antibacterial agent against bacteria causing mastitis revealed a similar positive response as antibiotics (penicillin and gentamycin) [18]. However, little information has been reported on CHT supplementation that focuses on performance in dairy cows and bacteria causing mastitis leading to high SCC in milk in crossbred dairy cows. Thus, we hypothesized that CHT can increase milk production and simultaneously reduce somatic cell counts. Therefore, our objective in this study was to determine the effects of HT extracts from sweet chestnut wood (*Castanea Sativa* Mill.) on intake, digestibility, rumen fermentation, milk performance and somatic cell counts in lactating crossbred dairy cows.

## 2. Materials and Methods

### 2.1. Animal Welfare Statement

The Animal Ethics Committee of Khon Kaen University approved the experimental protocol based on the National Research Council of Thailand’s Ethics of Animal Experimentation (Record no IACUC-KKU-86/2560).

### 2.2. Animals, Diets and Experimental Design

Four crossbred dairy cows (>75% Holstein Friesian), with an average of 5.50 ± 1.65 years, parity 2.00 ± 1.22, body weight of 467.60 ± 35.20 kg and 46.75 ± 25.72 days in milk (DIM), were randomly assigned to receive one of four levels of CHT supplementation using a 4 × 4 Latin square design. The experiment was performed in four periods of 21 days each. There was a 5-day washout period. The first 14 days were assigned for treatment adaptation, while the following 7 days were used for sample collection. All dairy cows were housed in individual crates (2 × 5 m^2^) on concrete floors with rubber stall mats and open-air ventilation at a dairy farm in the Department of Animal Science, Khon Kaen University, Thailand. Dietary treatments included a control treatment (no CHT supplementation) and CHT treatments with CHT supplementation at 3.15, 6.30 and 9.45 g CHT/day. Cows were individually offered concentrate and roughage based on their requirements for maintenance, milk production and gain [19], twice daily after milking times, at 7 am and at 3 pm. The concentrate was formulated using locally available feed ingredients (Table 1) and offered in the amount based on the ratio of concentrate to milk of 1:1.6. Two equal portions of concentrate were given at the milking in the morning and in the afternoon. Rice straw was offered ad libitum. At the time of the morning feeding, the CHT was on-top and then mixed thoroughly with concentrate. Clean water and mineral blocks were always available.

The CHT extract from sweet chestnut wood (*Castanea sativa* Mill.) was provided from the Animal Supplement & Pharmaceutical Co., Ltd., Pathum Thani, Thailand. The tannin content of chestnut extract was at least 73.5% tannin. The tannin is composed of 85% HT and other tannin in brown powdered forms (Globaphonol, Global Nutrition International, Fougères, France). HT is a complex mixture, including both gallotannins and ellagitannins.

### 2.3. Sampling Procedures and Analysis Methods

All cows were weighed on Day 1 and Day 21 of each period in the morning before feeding. The feed intake of concentrate and roughage were measured separately and recorded daily by weighing the amount offered and the refused feed during the morning feeding. The offered amount of rice straw was adjusted daily to ensure approximately 10% refusal after feeding. Samples of feed were collected during the last five days of each period and pooled for analysis. Faecal samples were collected from each dairy cow by rectal sampling at the same time during the last five days of each period and pooled for analysis. The feed and faeces samples were dried at 60 °C for 48 h before being ground through a 1 mm screen and analysed by the AOAC method [20] for ether extract (EE; method 920.39), ash (method 942.05) and crude protein (CP; method 984.13). Neutral detergent fibre (NDF) and acid detergent fibre (ADF) were measured using the detergent method [21]. Acid insoluble ash (AIA) was analysed as an internal marker to determine the apparent digestibility of nutrients [22].

On the last day of each period, 200 mL of rumen fluid samples were collected at 0, 1, 2 and 4 h post-feeding using a stomach tube connected to a vacuum pump. The rumen fluid samples were taken from the middle part of the rumen. The pH was immediately measured using a portable pH meter (CyberScan pH 11, Eutech Instruments Pte Ltd, Singapore). The rumen fluid samples were then filtered through four layers of cheesecloth. Samples were acidified by adding 5 mL of H_2_SO_4_ solution (1 M) mixed with 50 mL of rumen fluid to inhibit microbial activity. The mixture was then centrifuged at 16,000× *g* for 15 min, and the supernatant was stored at −20 °C. Ammonia nitrogen (NH_3_-N) was measured using Kjeldahl analysis (Kjeltec Auto 1030 Analyzer, Tecator, Hoganiis, Sweden), and volatile fatty acids (VFAs) were analysed using high-pressure liquid chromatography (HPLC model RF-10AXmugiL, Shimadzu, Tokyo, Japan) following Mathew et al. [23]. At the end of each period, blood samples were taken from the jugular vein immediately after the sampling of rumen fluid. Blood was drawn into 6 mL vacutainer tubes without anticoagulant, refrigerated at 5 °C for 1 h, and then centrifuged at 3000× *g* for 15 min. The supernatant was stored at –20 °C until analysis of the serum urea nitrogen (SUN) concentration using colorimetric method test kits and an automated analyser (Roche Diagnostics, Indianapolis, IN, USA) (COBAS INTE-GRA 400 plus analyzer, Roche Diagnostics, Indianapolis, IN, USA).

Cows were milked twice daily at 5 a.m. and 2 p.m. and recorded throughout the experiment. Milk samples were collected twice during each period. The first collection was on Day 13 (afternoon milking) and Day 14 (morning milking), and the second collection was on Day 20 (afternoon milking) and Day 21 (morning milking). The samples were pooled and stored at 4 °C until milk composition analyses. Fat, protein, lactose, total solids (TS), solids-not-fat (SNF) contents and SCC were analysed using MilkcoScan FT 600 and Fossomatic 500 Basic (Foss Electric, Integrated Milk Testing^TM^), respectively. Somatic cell score (SCS) was calculated as [log_2_(SCC × 10^−5^) + 3] [24]. Milk samples from each period were also collected to identify the bacteria that cause bovine mastitis, according to conventional biochemical tests [25,26].

### 2.4. Statistical Analysis

All data obtained from the experiment were subjected to ANOVA for a 4 × 4 Latin square design using general linear models (GLMs) procedures [27]. The model was:Y_ijkl_ = µ + P_i_ + C_j_ + T_k_ + e_ijkl_
where Y_ijkl_ is the observation, µ is the overall mean, P_i_ is the period effect (i = 4), C_j_ is the cow effect (j = 4), T_k_ is the treatment effect (k = 4) and e_ijkl_ is the residual error. The differences between means were compared using Tukey’s test. Orthogonal polynomial contrasts were performed to determine linear, quadratic and cubic responses to supplementing CHT levels. Somatic cell count values were transformed to log_10_ bases before the statistical analysis. The results were presented as least square means. Significance was declared at *p* ≤ 0.05, whereas tendencies were indicated at 0.05 ≤ *p* ≤ 0.10.

## 3. Results

### 3.1. Nutrient Composition of the Experimental Diets

The chemical composition of the feeds used in this study is presented in Table 1. The concentrate contained 90.73%, 21.68%, 3.20%, 19.14% and 14.40% of DM, CP, EE, NDF and ADF, respectively. The rice straw consisted of 92.83% DM, 3.21% CP, 1.80% EE, 90.63% NDF and 57.48% ADF. The metabolizable energy of the concentrate and rice straw was 3.18 and 1.36 Mcal/kg DM, respectively. The brown powder of chestnut hydrolysable tannin consisted of 91.72%, 9.63% and 0.75% of DM, CP and EE, respectively.

### 3.2. Feed Intake, Nutrient Intake and Nutrient Digestibility

Table 2 shows the effects of CHT supplementation on feed intake in lactating dairy cows. Rice straw intake decreased quadratically (*p* < 0.05) as the levels of CHT supplementation increased. However, the intake of concentrate and the total DM intake as kg/day, %BW and g/kg BW^0.75^ were not significantly different (*p* > 0.05) among the CHT treatments when compared with the control treatment.

The nutrient intake, apparent digestibility and energy intake are shown in Table 2. The nutrient intake of OM, EE, NDF and ADF did not differ (*p* > 0.05) between dietary treatments. The apparent digestibility of EE, NDF and ADF did not differ (*p* > 0.05) among the treatments. Dry matter (DM), OM, EE and ADF digestibilities in the CHT treatments were statistically higher (*p* < 0.05) than those in the control treatment. Crude protein digestibility in cows fed the CHT supplement at 9.45 g/day was significantly higher (*p* < 0.05) than that in cows fed the control. Moreover, the OM digestibility of diet was increased linearly (*p* < 0.05) as CHT supplementation increased, while OM digestibilities in the CHT treatments were higher (*p* < 0.05) than those in the control treatment.

The energy intake of cows ranged from 34.94 to 39.70 Mcal ME/day, which met their energy requirement for producing milk (15.03–17.16 kg/day). Cows supplemented with CHT consumed considerably more energy (*p* < 0.05) than cows without supplemented with CHT.

### 3.3. Rumen Fermentation Patterns and Blood Metabolites

Table 3 illustrates the effects of CHT supplementation on ruminal pH, ruminal NH_3_-N, total VFAs, acetic acid (C_2_), propionic acid (C_3_), butyric acid (C_4_), A:P ratio and serum urea nitrogen (SUN). Ruminal pH, ruminal NH_3_-N, acetic acid (C_2_), propionic acid (C_3_), butyric acid (C_4_) and the A:P ratio were not significantly different (*p* > 0.05) between the CHT treatments and the control treatment. The total VFA concentration increased linearly (*p* < 0.05) as the levels of CHT supplementation increased. In addition, SUN in cows linearly decreased (*p* < 0.01) as levels of CHT increased. Cows receiving 9.45 g of CHT/day had a lower SUN (*p* < 0.05) than in cows receiving 0 and 3.15 g of CHT/day, but it did not differ from cows receiving 6.30 g CHT/day.

### 3.4. Milk Yield and Milk Composition

Chestnut hydrolysable tannin supplementation had no effect on milk yield, 4% FCM, milk composition, fat and protein ratio, milk composition or nitrogen utilization efficiency (NUE) when compared with the control treatment (Table 4). The protein and lactose concentration in milk showed a linear increase (*p* < 0.05) with increasing CHT levels. Moreover, a linear tendency of lactose yield was observed (*p* = 0.09). Feed efficiency increased linearly (*p* < 0.05) as CHT supplementation increased. The feed efficiency of cows with the CHT treatments tended to linearly increase (*p* < 0.05) as CHT levels increased.

Somatic cell count and somatic cell score (SCS) were lower (*p* < 0.01) in CHT-supplemented cows than in the control. Cows receiving CHT supplementation had a linear decrease (*p* < 0.01) in SCC and SCS.

### 3.5. Identification of Mastitis-Causing Bacteria

Twenty isolates from 16 milk samples were identified as mastitis-causing bacteria, which were isolated and subjected to preliminary screening utilizing morphological and biochemical features. Six species of mastitis-causing bacteria were isolated from the milk collected in the experiment. In addition, one to three bacterial species were identified in a milk sample. The bacteria that were isolated were mostly from the environment. *Enterococcus* spp. were the most commonly found bacteria (35.0%) (Figure 1).

## 4. Discussion

### 4.1. Dry Matter Intake, Nutrient Intake and Nutrient Digestibility

The present study found that a higher level of CHT supplementation in the concentrate resulted in lower rice straw intake but not in a concentrated intake or overall DMI. This was most likely due to the astringent taste of tannins [31]. High tannin concentrations are linked to decreased feed intake, decreased feed palatability, decreased digestion and the development of conditioned aversion [32]. Kapp-Bitter et al. [33] found that supplementing with pellets containing 100 g/kg of chestnut extract resulted in significantly lower intake than supplementing with pellets containing 0 and 50 g/kg of chestnut. In addition, Aruerre et al. [34] reported that supplementation with mixed quebracho and chestnut tannin extract (2:1 ratio) at levels of 0, 0.45, 0.90 and 1.80% of dietary DM linearly reduced DM intake (25.5 to 23.4 kg/day). However, Liu et al. [11] found that feeding CHT at 1% of diet had no effect on the DMI of lactating cows. Another study [35] reported no effect on feed intake when CHT was included at 0.49% of dietary DM. Furthermore, mixing CHT into total mixed ration (TMR) had no effect on intake [6]. Although the effects of tannins on feed intake in ruminants remain controversial, it can be concluded that the level at which CHT has a negative effect may vary depending on the structures, sources and concentrations of tannins in the extract mixture.

Jayanegara et al. [36] reported that CHT is toxic to *R. flavefaciens* and anaerobic fungal populations, which could lead to decreased fibre digestion. However, the reduction in NDF and ADF digestion in this study was not observed. The digestibility of ADF was increased in the CHT groups compared to the control. In comparison to the control diet, Mannelli et al. [37] confirmed that the inclusion of CHT in the diet had no influence on NDF degradation. Due to the addition of CHT to the diet, the microbial community in the rumen was enriched, particularly the Firmicutes community, which included *Anaerovibrio, Bibersteinia, Escherichia/Shigella, Pseudobutyrivibrio* and *Streptococcus*. Deaville et al. [38] found that CHT is not bound to dietary fibre, which may explain why adding CHT from chestnut to the diet of lactating dairy cows in the study did not reduce NDF or ADF digestibility. In addition, Carrasco et al. [39] reported that supplementation with a mixture of chestnut and quebracho tannin at 0.2% in the diet enhanced fibrolytic, amylolytic and ureolytic bacterial communities in the rumen and reduced methanogenic archaea. Another study [36] also reported similar findings in that hydrolysable tannins had a lower effect on the degradation of nutrients by rumen microbes than condensed tannins. In this trial, cows given CHT showed an increase in the digestibility of DM, OM and CP. In contrast, Aguerre et al. [35] found a linear decrease in DM, OM and CP digestibility when tannins from quebracho-chestnut tannin extracts increased from 0.45 to 1.80% of DM. In addition, one study [40] also showed a dramatic decrease in DM, CP, NDF and ADF digestibility in cows fed a 3% valonia hydrolysable tannin diet. According to the literature, the extent and sources of tannin have an impact on intake and feed utilization in ruminants.

### 4.2. Rumen Fermentation Patterns and Blood Metabolites

Ruminal pH was not altered by CHT treatments, which varied from 6.81–6.98. The optimum ruminal pH level for the microbial digestion of fibre and protein, according to Calabrò et al. [41], and Wanapat and Cherdthong [42], is 6.5 to 7.0. Furthermore, our results are similar to those of a previously published study [43], which found that ruminal pH was 6.79 with CHT of chestnut tannin with oil supplemented at 100 g/kg of DM in the diets of rumen-fistulated sheep. This was also reported in the study of Herremans et al. [44], which found that supplementation with hydrolysable tannins from oak (26 g/kg DM) resulted in a ruminal pH of 7.0, with no difference between diets.

The results of this study revealed that CHT supplementation had no effect on ruminal NH_3_-N levels that were within a normal range [45]. The ruminal NH_3_-N concentration, on the other hand, varied depending on several factors, including rumen protein degradation, feed protein amount and retention time [46]. Moreover, Aguerre et al. [34] reported that the tannin extract mixture from quebracho and chestnut trees (2:1 ratio) supplemented at levels of 0.45, 0.90 and 1.80% of diets reduced the concentration of NH_3_-N in ruminal fluid. The outcome was a decrease in rumen protein degradation due to the formation of tannin–protein complexes. Ali et al. [12] confirmed that CHT supplementation in a diet prevents the degradation of protein in the rumen and consequently increases the supply of amino acids in the intestines. These results are in line with earlier findings [47,48] indicating that reduced protein degradation and production of NH_3_-N in the rumen resulted in decreased absorption of ammonia across the rumen wall into the blood. The results of this study indicate that SUN concentrations have similarities to the normal range from 8 to 20 mg/dL [49]. However, the SUN concentration in this experiment showed a quadratic decrease with increasing levels of CHT. This finding was in line with results from a previously published report [34]. Serum urea nitrogen decreased linearly when the tannin mixture content in the diet increased from 0 to 1.80% of DM. Other reports have shown a similar response to condensed tannin supplementation that reflects lower rumen protein degradation [50].

Total VFA concentrations increased linearly with increasing CHT levels. It is postulated that CHT supplementation reduces roughage intake and leads to an increased ratio of concentrate to roughage in the diet. A high proportion of concentrate in the ration led to an increase in VFA production [51,52]. This result agrees with previously published work [53], in which ewes fed CHT from chestnut had a higher total VFA level than ewes fed condensed tannin and the control ewes (73.07, 39.82 and 50.89 mM, respectively). Costa et al. [44] reported that sheep fed CHT had a higher total VFA concentration (152.2 mM) than sheep fed condensed tannin (79.3 mM) (*p* < 0.001). In contrast, Bhatta et al. [54] reported that increasing tannin levels led to a reduction in total VFA concentrations. Several studies [55,56,57] have shown that tannin did not influence the total VFA concentration or their molar proportions in rumen fluid from ewes. Other authors have shown that a response to high levels of tannin (>1.0% of DM) added to the diet decreases the total VFA concentration [35,50,58]. In addition, Benchaar et al. [59] reported that total VFA concentrations were not affected by feeding quebracho tannin extracts at 0.64% of DM.

### 4.3. Milk Yield and Milk Composition

Supplementation with CHT had no effect on milk yield, 4% FCM, energy-corrected milk (ECM), fat- and protein-corrected milk (FPCM), milk composition or nitrogen utilization efficiency. Consistent with our results, a previous study found that tannin supplementation did not affect milk yield; 4% FCM; energy-corrected milk (ECM); or fat, protein and lactose contents [50]. Sliwinski et al. [34] found no effect of CHT on feed intake and milk production when CHT from chestnut tannin was included at 0.49% of dietary DM. Likewise, another study [10] observed no significant difference in milk production or milk fat when cows received diets with CHT at 0.30% chestnut tannin extract. Liu et al. [11] observed that HTs from chestnut tannin in diets did not affect milk yield, 3.5% FCM, milk protein, milk fat, or lactose in milk. In another study [60], milk yield was not affected by offering silage treated with tannin. According to Zhang et al. [40], the supplementation of lactating cows with 3% valonia hydrolysed tannin had no effect on milk yield, FCM, ECM, or the composition of fat, protein and lactose in the milk. This was in line with the results of another study [45] that reported that no significant differences between treatments were detected in milk yield and major milk components compared with the control group when CHT from oak was included at 26 g/kg of DM in the forage of dairy cows. According to Dubey [61], a significant increase in milk yield from 9.44 to 10.35 kg/day/cow was achieved by adding 3% tannin from *Acacia nilotica* pods in crossbred cows. Additionally, Colombini et al. [60] reported slight increases in fat-corrected milk (*p* < 0.10), and the supplementation of chestnut tannin (120 g/day) in cows’ diets was found to increase milk yield [62].

The results of this study revealed that SCC and SCS in dairy cows fed with CHT were significantly different compared to the control. Ali et al. [12] reported that CHT supplementation ranging from 20 to 40 g/day reduced SCC in lactating crossbred cows. The authors suggested that the lower SCC in milk may be due to the inclusion of zinc. Zinc plays a role in optimizing the cellular immune response [63] and the formation of keratin, which entraps these bacteria [64]. Furthermore, Liu et al. [11] reported that milk SCS in postpartum cows fed a diet containing chestnut tannin (a species with high CHT) at 1% of dietary DM decreased (*p* < 0.05) when compared to the control group (2.37 and 3.78, respectively). Liu et al. [65] found that HTs from *M. bracteata* have a strong anti-inflammatory effect, which could reduce SCC in milk. Based on these results, it may be concluded that CHT is able to act against bovine mastitis pathogens because it has antimicrobial properties [18,66]. Due to its absorption and transfer rate to the udder, the extent and source of CHT supplementation are important. Recently, it was reported that the transfer of a phenolic compound at high levels in CTs or CHT from various tanniferous forages in the diet to milk was low, ranging between 1 and 2% [67]. To reduce SCC or SCS in milk, further research should focus on the effective transport of phenolic compounds, particularly hydrolysable tannins, from the diet to the udder.

### 4.4. Identification of Mastitis-Causing Bacteria

In this experiment, CHT supplementation had positive effects against mastitis-causing bacteria. The prevalent bacteria isolated that were related to causing mastitis were environmental bacteria, and contagious bacteria were not identified. Türkyilmaz et al. [68] reported that environmental microorganisms (77.1%) were found in 188 isolated milk samples, which was much higher than the level of contagious microorganisms found (22.9%). Another study [69] indicated that low production rates and high SCC are usually observed in *S. agalactiae* infections. Upon infection by *S. agalactiae*, the SCC in milk was extremely high (>1,572,000 cells/mL) [17]. Furthermore, Panneum et al. [70] found that environmental pathogens such as coliforms, environmental *Staphylococci* spp., and environmental *Streptococci* spp. were the bacteria most often isolated in milk with a high SCC concentration. According to Juangphanich et al. [71], the incidence of subclinical mastitis in crossbred Holstein Friesian dairy cows in the central parts of Thailand is high, and environmental infections are the main reason.

Higher SCC levels indicate high intramammary inflection by mastitis-causing pathogens, which is associated with decreased lactose production and lower milk yield. In this study, cows with a high milk yield were observed to have a lower SCC in their milk. Subclinical mastitis affects the alveoli in the mammary, and mastitis bacteria-induced inflammation causes milk production to decrease. Hagnestam-Nielsen et al. [72] reported that daily milk loss at an SCC of 500,000 cells/mL ranged from 0.7 to 2.0 kg (3 to 9%) in primiparous cows, while in multiparous cows, the corresponding loss was 1.1 to 3.7 kg (4 to 18%), depending on the stage of lactation.

## 5. Conclusions

Although supplementation with CHT had no effect on feed intake or milk production and composition in this study, it did have a positive effect on nutrient digestibility as well as higher total VFAs and lower SUN in cows supplemented with CHT at 9.45 g/day. SCC in milk from cows infected with environmental bacteria was reduced by CHT supplementation. Long-term CHT supplementation in lactating dairy cows is also recommended to improve milk production and control mastitis.

## Figures and Tables

**Figure 1 vetsci-10-00269-f001:**
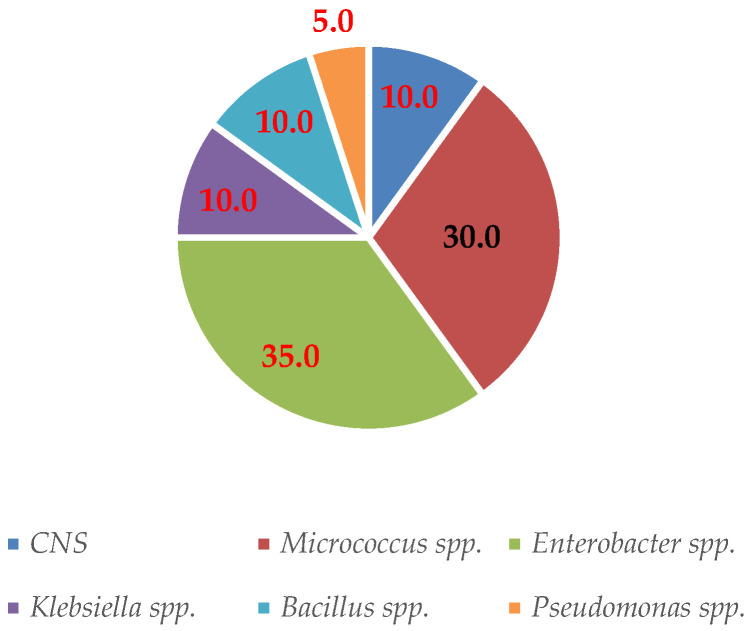
Distribution of isolated bacterial species (%) found in milk.

**Table 1 vetsci-10-00269-t001:** Ingredients of the concentrate and chemical composition of concentrate, rice straw and chestnut hydrolysable tannin extract powder.

Item	Concentrate	Rice Straw	Chestnut Hydrolysable Tannin Extract Powder
** *Ingredients, %DM* **			
Cassava chip	47.00		
Corn meal	7.00		
Rice bran	5.00		
Soybean meal	20.00		
Peanut hulls	4.00		
Palm kernel meal	9.50		
Residue sugar	3.00		
Urea	2.50		
Dicalcium phosphate	1.00		
Salt	0.50		
Premix	0.50		
Total	100.00		
***Chemical composition*** (*%DM*)			
Dry matter, %	90.73	92.83	91.72
OM	93.32	88.13	97.75
CP	21.68	3.21	9.63
EE	3.20	1.80	0.75
Ash	6.68	11.87	2.25
NDF	19.14	90.63	-
ADF	14.40	57.48	-
ME, Mcal/kgDM	2.62	1.53	-

OM = organic matter; CP = crude protein; EE = ether extract; NDF = neutral detergent fibre; ADF = acid detergent fibre; metabolizable energy (ME) calculated from the energy of each ingredient.

**Table 2 vetsci-10-00269-t002:** Nutrient intake and digestibility in lactating dairy cows supplemented with various levels of chestnut hydrolysable tannin.

Items	Chestnut Hydrolysable Tannin, g/day				SEM ^c^	*p* Value ^d^			
	0	3.15	6.30	9.45		Control vs. CHT	L	Q	C
Avg. BW, kg	458.4	464.6	480.9	481.4	10.54	0.21	0.12	0.79	0.60
DM intake,									
Concentrate	11.0	10.8	11.7	12.0	0.51	0.39	0.12	0.63	0.47
Rice straw	5.1 ^ab^	6.0 ^a^	5.4 ^ab^	4.7 ^b^	0.27	0.39	0.18	*	0.31
Total, kg/day	16.0	16.8	17.1	16.7	0.72	0.35	0.49	0.44	0.90
%BW	3.49	3.60	3.56	3.47	0.09	0.62	0.80	0.29	0.83
g/kg W^0.75^	161.5	167.2	166.8	162.4	4.81	0.50	0.92	0.33	0.93
Nutrient intake, kg/day									
OM	14.6	15.2	15.7	15.3	0.64	0.33	0.44	0.44	0.79
CP	3.1	3.0	3.3	3.3	0.14	0.36	0.13	0.78	0.50
EE	0.5	0.5	0.5	0.5	0.03	0.70	0.42	0.62	0.89
NDF	7.0	1.6	1.2	6.8	0.38	0.63	0.58	0.22	0.52
ADF	4.4	5.2	4.6	4.6	0.25	0.17	0.93	0.13	0.12
Apparent digestibility, %									
DM	68.1 ^a^	70.6 ^ab^	72.7 ^ab^	75.8 ^b^	1.35	*	**	0.81	0.87
OM	71.6 ^a^	73.4 ^a^	75.4 ^ab^	78.5 ^b^	1.44	*	*	0.68	0.91
CP	77.7 ^a^	81.0 ^ab^	81.2 ^ab^	84.6 ^b^	1.76	0.07	*	0.97	0.44
EE	81.0 ^a^	83.3 ^ab^	91.1 ^a^	85.3 ^ab^	1.81	*	*	0.07	0.06
NDF	47.3	50.6	48.1	55.8	2.16	0.29	0.15	0.51	0.30
ADF	37.5 ^a^	44.9 ^ab^	40.9 ^a^	53.2 ^b^	3.18	*	*	0.47	0.10
Energy intake ^1^									
Mcal ME/day	42.2	41.9	45.2	44.2	1.88	0.49	0.31	0.84	0.39

^1^ 1 kg of digestible organic matter (DOM) = 3.8 Mcal ME [28]. ^a,b^ Means with different superscripts in the same row differ (*p* < 0.05), * *p* < 0.05, ** *p* < 0.01. ^c^ Standard errors of the mean. ^d^ Probability of linear (L), quadratic (Q) and cubic (C) effects of chestnut hydrolysable tannin levels.

**Table 3 vetsci-10-00269-t003:** Ruminal pH, NH_3_-N, volatile fatty acids (VFAs) and SUN in lactating dairy cows supplemented with various levels of chestnut hydrolysable tannin.

Items	Chestnut Hydrolysable Tannin, g/day				SEM ^c^	*p* Value ^d^			
	0	3.15	6.30	9.45		Control vs. CHT	L	Q	C
Rumen end-products									
Ruminal pH	6.92 ^ab^	6.98 ^ab^	6.81 ^b^	6.87 ^ab^	0.03	0.37	*	0.93	0.01
Ruminal NH_3_-N, mg/dL	13.6	16.4	17.8	13.1	1.67	0.30	0.99	0.06	0.54
Total VFAs, mM	89.3	99.6	102.8	105.1	4.62	*	*	0.42	0.77
VFA profiles, mol/100 mol									
Acetic acid (C2)	64.7	66.7	67.2	65.4	1.14	0.22	0.61	0.14	0.89
Propionic acid (C3)	22.5	21.6	21.3	23.0	0.77	0.55	0.77	0.13	0.65
Butyric acid (C4)	12.8	11.7	11.6	11.6	0.58	0.12	0.20	0.34	0.75
A:P ratio	2.90	3.13	3.18	2.99	0.90	0.22	0.57	0.15	0.90
Blood metabolites									
SUN, mg/dL	21.8 ^a^	21.1 ^a^	20.5 ^ab^	15.4 ^b^	1.16	0.09	**	0.10	0.41

^a,b^ Means with different superscripts in the same row differ (*p* < 0.05), * *p* < 0.05, ** *p* < 0.01. ^c^ Standard errors of the mean. ^d^ Probability of linear (L), quadratic (Q) and cubic (C) effects of chestnut hydrolysable tannin levels. SUN = serum urea nitrogen.

**Table 4 vetsci-10-00269-t004:** Milk yield and milk composition in lactating dairy cows supplemented with various levels of chestnut hydrolysable tannin.

Items	Chestnut Hydrolysable Tannin, g/day				SEM ^c^	*p* Value ^d^			
	0	3.15	6.30	9.45		Control vs. CHT	L	Q	C
Milk yield, kg/day	18.2	17.9	19.9	20.5	1.17	0.34	0.13	0.73	0.50
4% FCM ^1^, kg/day	19.0	19.5	20.8	21.6	1.47	0.38	0.21	0.91	0.84
Milk composition									
Fat, %	4.35	4.61	4.27	4.48	0.22	0.69	0.96	0.90	0.28
Protein, %	3.50	3.45	3.25	3.36	0.05	0.03	0.02	0.12	0.07
Lactose, %	4.89	4.94	5.27	5.16	0.09	0.06	0.02	0.38	0.11
SNF, %	9.08	9.08	9.23	9.22	0.06	0.22	0.08	0.92	0.29
TS, %	13.43	13.69	13.50	13.69	0.25	0.52	0.62	0.90	0.49
Composition yield									
Fat, kg/d	0.79	0.82	0.85	0.91	0.07	0.43	0.28	0.90	0.95
Protein, kg/day	0.63	0.62	0.65	0.69	0.04	0.60	0.23	0.53	0.77
Lactose, kg/day	0.90	0.89	1.05	1.06	0.07	0.26	0.09	0.91	0.36
SNF, kg/d	1.65	1.63	1.84	1.89	0.11	0.36	0.12	0.75	0.46
TS, kg/d	2.44	2.45	2.69	2.80	0.18	0.36	0.15	0.81	0.67
ECM, kg	18.9	19.2	20.8	21.6	1.44	0.38	0.19	0.87	0.77
FPCM ^2^, kg	19.1	19.3	20.6	21.5	1.38	0.41	0.22	0.83	0.85
Fat:Protein ratio	1.25	1.34	1.31	1.33	0.06	0.29	0.44	0.54	0.58
Feed efficiency	1.12	1.06	1.15	1.22	0.03	0.42	0.02	0.07	0.22
NUE ^3^	0.20	0.20	0.19	0.20	0.004	0.40	0.69	0.17	0.26
SCC, log_10_	5.66 ^a^	5.17 ^b^	5.09 ^b^	5.13 ^b^	0.06	**	**	**	0.33
SCS ^4^	5.18 ^a^	3.56 ^b^	3.30 ^b^	3.42 ^b^	0.21	**	**	**	0.35

4% FCM = 4% fat-corrected milk, SNF = solid not fat, TS = total solid, SCC = somatic cell count. ^1^ 4% FCM = 0.4 × milk yield (kg/d) + 15 × fat yield (kg/d). ^2^ Fat- and protein-corrected milk (FPCM) = milk (kg/d) × [0.1226 × fat (%) + 0.0776 × protein (%) + 0.2534] [29]. ^3^ Nitrogen utilization efficiency = (milk protein yield (kg/d) ÷ 6.38)/(crude protein intake. (kg/d) ÷ 6.25) [30]. ^4^ Somatic cell score (SCS) = log_2_ (SCC/100,000) + 3 [24]. ^a,b^ Means with different superscripts in the same row differ (*p* < 0.05), ** *p* < 0.01. ^c^ Standard errors of the mean. ^d^ Probability of linear (L), quadratic (Q) and cubic (C) effects of chestnut hydrolysable tannin levels.

## Data Availability

The data that support the findings of this study are available from the corresponding author, C.J., upon reasonable request.

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
