# Peer review of "Effects of Chestnut Hydrolysable Tannin on Intake, Digestibility, Rumen Fermentation, Milk Production and Somatic Cell Count in Crossbred Dairy Cows"

_vetsci, 2023, doi:10.3390/vetsci10040269_

Round 1

Reviewer 1 Report

The current manuscript aims to determine the effects of chestnut hydrolysable tannin on intake, digestibility, rumen fermentation, and milk production and quality of crossbred dairy cows. The subject is of average scientific soundness. The simple summary should be changed to be simpler and more explicit. The used methods should be more explained; authors most of the time, refer to other works without mentioning the methods briefly. The statistical method mentioned in the material and methods section is not corresponding to the presented results. Also, the short duration of the experiment does not provide reliable results.

There are some comments

Simple summary:

This summary should be simplified more. The experimental design is unnecessary, and the abbreviation should be introduced.

Line 29-31: the sentence should be simplified.

Abstract:

The methodology and studied parameters did not mention

Introduction:

Line 46: this information should be verified or a reference should be added, because tannins are considered an anti-nutritional factor that limits nitrogen digestibility and reduces intake, etc.

Material and methods

The ages of cows and the number of pregnancies should be added

Why did the authors choose these doses 3.15, 6.30, and 9.40 g CHT/day?

Section 2.3.: please add a subsection for each type of analysis

The used analysis method should be more detailed.

Statistical analysis: cows were considered as a random factor, so it is not an ANOVA, it is a PROCMIXED test.

In the text, authors talk about linear decrease, the statistical model that should be used is the orthogonal contrast, not an ANOVA

Results:

Section 3.1. should be reported in material and methods, not as results.

Line 223 -225: the reported parameters did not mention in the Material and methods section

Line 226: verify the probability

Line 228-229: check the sentence and probabilities

Discussion

Authors should clarify which type of tannins they are talking about in the discussion

Line 278: TMR , not defined before?

Line 309: names of authors of references 42 and 43 should be added

Conclusion:

Authors should apport the practical use and the interest of the finding.

Reviewer 2 Report

This is a relevant topic. I have attached detailed comments. Despite the interest in the topic, science should be repeatable. The named product that was tested is not available worldwide, not found on the producers home page and it is imperative that you report the purity of hydrolysable tannins (or mixture). 

The methods section needs revision. The methods to determine microbial protein and total protein digestibility are missing. 

There is also a serious misunderstanding (misrepresentation?) in line  284-285. Fiber digestion is reported through NDF and ADF apparent digestibility. Finally, in the discussion, a serious problem arises in the misunderstanding of chestnut and oak tannins. 

Round 2

Reviewer 1 Report

The manuscript has been improved. However, the authors did not explain why they did choose the level of CHT incorporation, and to use the orthogonal contrast the level of incorporation should be a multiplication of the 1st dose. Below you find some comments.

Line 51: “Hydrolysable or condensed tannins », please replace “or” by “and”

Line 69: use the abbreviation “HT”

Line 91: please add the SE for the average of age and parity.

Line 99: authors did not answer the question of why they did use these concentrations of CHT? And why they did not supplement the last group by 9.45%

Line 110: If I understand, authors used an extract that contains CT and HT. The observed effect could be attributed to CT also.

Line 111: Please use abbreviations.

Line 160: to use the contrast, authors should use 9.45% (3.15% x 3) of CHT

The conclusion should be improved to demonstrate the potential use of these found.

Line 436: please remove “.” before “Has been”

Reviewer 2 Report

I still think it is relevant to test commercial products. However, my primary consideration is that the effects seen by the addition of the CHT mixture can not be solely attributed to the tannins. I strongly suggest a careful revision. The authors have not tested the carrier material.  Either the manufacturer must declare what the carrier material is or the authors must undertake a careful revision throughout the text where the results are  attributed the "commercial CHT product" or "CHT mixture" 

line 31: "tended to quadratically decrease rice straw intake (P < 0.05)". if P<0.05 it is significant (not a tendency)

Line 53-54 Please revise for correct English. I SUGGEST:  "whereas hydrolysable tannins (HTs) have a lower molecular weight than CTs and are more easily absorbed from the intestine, potentially increasing their toxicity to animals.

line 65 condition (not conditional) 

line 107 "At the time of the morning feeding, the CHT was replenished with concentrate." this indicates that the CHT mixture (powder) was in the feed crib all night was refilled with concentrate. This could be a bit clearer. 

110-112 Thank you for providing the protein, fiber and ether extract value of the "Chestnut hydrolysable tannin extract powder". The title in this table tells me that this analysis is for the powder with tannins. However, in addition to the tannins this powder contains 26.5% something else. It is unclear to me if the "brown powder is an inert carrier substance or has antibiotic properties on its own. I am uncertain if you have tested the "brown powder" as this could contain an ingredient that could be  the reason for the results. 

Line 171 suggests that the carrier material for the tanins was tested alone. PLEASE clarify this. 

line 421: "In this study, it was discovered that cows with a high milk yield had a lower SCC in  their milk." PLEASE modify this. - The greater the yield, the greater the dilution effect. This sentence needs to acknowledge common knowledge that SCC is not only affected by breed, parity, stage of lactation, and commercial additive but also most certainly by yield. 
